# Robot Arm Reaching Based on Inner Rehearsal

**DOI:** 10.3390/biomimetics8060491

**Published:** 2023-10-18

**Authors:** Jiawen Wang, Yudi Zou, Yaoyao Wei, Mengxi Nie, Tianlin Liu, Dingsheng Luo

**Affiliations:** 1National Key Laboratory of General Artificial Intelligence, Key Laboratory of Machine Perception (MoE), School of Intelligence Science and Technology, Peking University, Beijing 100871, China; chiawenw@stu.pku.edu.cn (J.W.); zyudi@pku.edu.cn (Y.Z.); yaoyaowei@pku.edu.cn (Y.W.); niemengxi@pku.edu.cn (M.N.); liutl@pku.edu.cn (T.L.); 2PKU-WUHAN Institute for Artificial Intelligence, Wuhan 430073, China

**Keywords:** arm reaching, motion planning, inner rehearsal, internal model, human cognitive mechanism

## Abstract

Robot arm motion control is a fundamental aspect of robot capabilities, with arm reaching ability serving as the foundation for complex arm manipulation tasks. However, traditional inverse kinematics-based methods for robot arm reaching struggle to cope with the increasing complexity and diversity of robot environments, as they heavily rely on the accuracy of physical models. In this paper, we introduce an innovative approach to robot arm motion control, inspired by the cognitive mechanism of inner rehearsal observed in humans. The core concept revolves around the robot’s ability to predict or evaluate the outcomes of motion commands before execution. This approach enhances the learning efficiency of models and reduces the mechanical wear on robots caused by excessive physical executions. We conduct experiments using the Baxter robot in simulation and the humanoid robot PKU-HR6.0 II in a real environment to demonstrate the effectiveness and efficiency of our proposed approach for robot arm reaching across different platforms. The internal models converge quickly and the average error distance between the target and the end-effector on the two platforms is reduced by 80% and 38%, respectively.

## 1. Introduction

In recent years, robots have played important roles in many fields, especially for humanoid robots. As arm manipulation is one of the most basic abilities for human beings [1], arm motion control is also an indispensable ability for humanoid robots [2]. In modern factories, automation manufacturing and many other production activities are inseparable from robot arms [3]. Among various types of arm manipulation abilities, arm reaching is one of the most basic, and it is the first step in many complex arm motions, such as grasping and placing, and can also lay the foundation for subsequent motion, perception, and cognition [4,5].

The major goal of robot arm reaching is to choose a set of appropriate arm joint angles so that the end-effectors can reach the target position in Cartesian space with a certain posture, which is commonly referred to as internal model control (IMC) [6]. In 1998, Wolpert et al. [7] reviewed the necessity of such an internal model and the evidence in a neural loop. In the same year, Wolpert and Kawato [8] proposed a new architecture based on multiple pairs of inverse (controller) and forward (predictor) models, where the inverse and forward models are tightly coupled. We usually implement robots’ motion control through kinematic or dynamic modeling [9]. However, since the control of force and torque is not involved in robot arm reaching tasks, only the kinematic method is considered. The internal kinematics model can be separated into the inverse kinematics (IK) model and forward kinematics (FK) model according to the input and output of the model.

In the task of reaching, the major problem is how to build an accurate IK model that maps a certain posture to a set of joint angles. There are mainly two types of approaches to robot arm reaching control: conventional IK-based approaches and learning-based approaches. Table 1 shows some related research.

In conventional IK-based approaches, the mapping from the pose of the end-effector to the joint angles is built based on the mechanical structure of the robot’s arm. The joint angles are calculated with an analytical method or iterative method [19]. In this type of approach, the accuracy of the IK model strongly depends on the measurement accuracy of the robot’s mechanical parameters, and we need to solve equations in high dimensions [20]. This brings difficulties in calibrating the parameters, which may change continuously because of the wear and tear of the robot. The application of conventional IK-based approaches in complex and unstructured environments is limited by the accuracy of the measurement and the manual calibration of the parameters.

To avoid the drawbacks of conventional IK-based approaches, researchers have focused on learning methodologies to manipulate robot arms, rather than control-theoretic approaches, in recent years.

It is recognized that robots must have the ability to learn continuously in order to adapt to the ever-changing and unstructured environment. Learning can also decrease the complexity of control for robots with large degrees of freedom [21]. The learning-based approaches are inspired by cognitive, motion, and other relevant mechanisms in human beings. The inverse model is established by means of self-exploration, based on neural networks [22], reinforcement learning [18], or other learning algorithms.

Using a learning-based model, accurate measurements of the robot’s mechanical parameters are no longer the decisive factor in arm reaching. However, a well-performing model requires a large amount of training data, mainly generated from trial-and-error experiments, which might cause great abrasion to the robot. Many researchers train the model in simulation first and then refine the model in a real robot platform [23].

To perform well in robot arm reaching tasks, it is necessary to ensure the accuracy of the internal model and the target positioning accuracy as well. As discussed above, in arm reaching manipulation, the joint angles can be calculated by the inverse kinematics model in IK-based approaches or the inverse model built through learning-based approaches once we know the position and posture of the end-effector, while the target pose is mainly determined through visual positioning, which is strongly related to the performance of the camera.

Compared with the absolute positioning method, relative positioning will help to reduce the influence of perception error. Research also shows that in the process of human arm reaching, older children and adults consider both target and hand positions [24,25]. Based on this mechanism, Luo et al. [26] proposed a relative-location-based approximation approach in 2018. In their work, the reaching process is divided into two stages, rough reaching and iterative adjustment. However, the motion commands are combined with six basic moving directions, which may lead to non-smooth reaching trajectories. To smooth the trajectory of the reaching trajectory, we use differential theory for reference and limit the distance of each movement to a small-scale given threshold.

The reaching approaches described in the above research are mainly in an open-loop manner [27,28]. Each joint of the robot arm moves to a target angle calculated by the inverse model, where feedbacks are less considered. However, when it comes to the real environment, the planning becomes more difficult and less robust [29], and the motion commands generated by the inverse model may not be executed precisely because of mechanical errors. Thus, feedbacks on the execution and corresponding motion planning are needed to increase the reaching performance. In 2017, Krivic et al. [30] proposed a planning approach to predict new information about a partially known state and to decrease uncertainty. In 2019, Luo et al. [31] proposed an action selection method based on sensorimotor prediction and anticipating the consequences of currently executable actions in internal simulation.

In this paper, we propose an inner-rehearsal-based robot arm reaching approach, in which IK models are obtained by the learning method and adjusted with closed-loop feedback. Our intention was to explore an alternative approach to robotic arm reaching that can enhance the learning efficiency of models and reduce the dependence on model configuration. Firstly, a staged learning method to establish the internal models is proposed, with the help of a coarse forward kinematics model. In the first stage, the internal models are trained with data generated from the FK model. Then, in the second stage, the internal models are fine-tuned with feedback from actual interactions with the external environment. The loss of the inverse models during training is defined as the distance between the target and the predicted position in Cartesian space calculated by the forward model. After the internal models are well trained, a motion planning method based on inner rehearsal is proposed. Robots can predict the result of a motion command without actually executing it through inner rehearsal [32]. Based on the prediction of the current motion command, the robot can generate the next motion command. In this way, a sequence of motion commands will be generated based on inner rehearsal, which could lead the end-effector toward the target. Our preliminary work has proven the effectiveness of inner rehearsal [14].

Experiments on the Baxter robot in simulation and the humanoid robot PKU-HR6.0 II in a real environment show the effectiveness and celerity of the proposed approach to robot arm reaching on different platforms.

The main contributions of this paper are as follows.

The internal models are established based on the relative positioning method. We limit the output of the inverse model to a small-scale displacement toward the target to smooth the reaching trajectory. The loss of the inverse model during training is defined as the distance in Cartesian space calculated by the forward model.The models are pre-trained with an FK model and then fine-tuned in a real environment. The approach not only increases the learning efficiency of the internal models but also decreases the mechanical wear and tear of the robots.The motion planning approach based on inner rehearsal improves the reaching performance via predictions of the motion command. During the whole reaching process, the planning procedure is divided into two stages, proprioception-based rough reaching planning and visual-feedback-based iterative adjustment planning.

## 2. Related Work

This section describes previous studies on visual servoing reaching, internal model establishment, and inner rehearsal.

### 2.1. Reaching with Visual Servoing

The general aim of the visual servo system in reaching tasks is to reduce the error e(t) [33], defined as
(1)$$e\left(t\right)=s\left(m\left(t\right),a\right)-s^{*}$$
where m(t) represents image measurements, *a* indicates any potential additional data, and $s^{*}$ stores the desired final position.

As an effective method, visual servoing has been applied in motion estimation [34], position control [35], and other robotics tasks. In 2000, Shen et al. proved that with visual servoing, the robot trajectory can approach the desired trajectory asymptotically [35]. In 2013, Lampe et al. proposed a reinforcement learning system for autonomous robot arm reaching using visual servoing. They used a visual feedback control loop to realize control, making it both reactive and robust to noise [17].

### 2.2. Learning-Based Internal Model

In 1987, Kosuge et al. introduced the concept of a virtual internal model and applied it in robot arm control [36]. In 1993, Koivisto et al. analyzed a nonlinear internal model controller realized with a multilayer perceptron neural network and proved that implementing practical experience with neural networks can provide IMC robust performance [37], demonstrating that a learning-based internal model is reliable.

In recent years, the learning-based internal model has been used in ethical robots [38], robot intent prediction [39], robot manipulation [18], and other robotics research; it simulates the cognitive mechanisms of humans and makes robots more intelligent.

### 2.3. Inner Rehearsal

Humans can simulate the normal execution results of an action through inner rehearsal [40]. Taking advantage of the inner rehearsal mechanism, people can try to run the potential actions in their minds and predict their results, such as actions or decisions that are not explicitly executed [41,42].

For robots, through inner rehearsal, the result of a motion command can be predicted without actual execution [32]. Inner rehearsal has been used in robot linguistic interpretation [43], relation learning [44], and navigation [45] to predict the result of a command and choose an action accordingly. In this way, robots can avoid unnecessary attempts and conduct the best move. In their recent work, Atkinson et al. proposed to use pseudo-rehearsal in reinforcement learning to prevent neural networks from catastrophic forgetting [46,47].

### 2.4. Issues Associated with Related Work

To avoid the drawbacks of the conventional IK-based approach, we implement the following.

We use image-based visual servoing to construct a closed-loop control so that the reaching process can be more robust than that without visual information.We build refined internal models for robots using deep neural networks. After coarse IK-based models generate commands, we adjust the commands with learning-based models to eliminate the influence of potential measurement errors.Inner rehearsal is applied before the commands are actually executed. The original commands are adjusted and then executed according to the result of inner rehearsal.

## 3. Methodology

This section describes the proposed robot arm reaching approach in detail. The overall framework, the establishment of the internal models, and the inner-rehearsal-based motion planning approach are introduced.

### 3.1. Overall Framework

The overall framework of the proposed approach is shown in Figure 1. The proposed method comprises four blocks: (1) visual information processing, (2) target-driven planning, (3) inner rehearsal, and (4) command execution.

The target position in Cartesian space is generated after the robot sees the target object through the visual perception module. The visual stimulation is converted into the required visual information, and then the intrinsic motivation is stimulated to generate the target [1,48].The aim of movement is generated by the relative position between the target and the end-effector. The inverse model generates the motion command based on the current arm state and the expected movement. Each movement is supposed to be a small-scale displacement of the end-effector toward the target.The forward model will predict the result of the motion command without actual execution. The predictions of the current movement are considered to be the next state of the robot so that the robot can generate the next motion command accordingly. In this way, a sequence of motion commands will be generated. The robot conducts (2) and (3) repeatedly until the prediction of movements exactly reflects the target.The robot executes these commands and reaches the target.

### 3.2. Establishment of Internal Models

The internal models are used to form the mapping between the joint angles and the body state, which consists of the forward model and the inverse model. The inverse model is used to generate a corresponding motion command based on the expected movement, while the forward model is used to predict the end-effector’s position based on the given joint angles, and thus to evaluate the effect of the command.

The internal models are trained through two stages. In the first stage, the models are pre-trained using a coarse forward kinematics model where the robot realizes self-exploration through motor babbling [49]. In this stage, all work is done in simulation and no actual movement is conducted. This allows the model to gain a basic understanding of its own capabilities and the interaction with the environment. The coarse forward kinematics model is more in line with the reality of the situation. Because forward models can also generate errors in practical environments, we achieve high-precision control by drawing on human cognitive mechanisms. This is what we emphasize in this work. In the second stage, the internal models are optimized with sensorimotor data collected from the real execution of the robot. The inverse model is further trained with visual feedback. The two-stage learning approach of the internal models reduces the amount of training data required from real robots and thus reduces the mechanical wear and tear of the robot.

#### 3.2.1. Modeling of the Internal Models

The internal model is a controller, described by Formula (Equation 2). In Formula (Equation 2), a motion command Δq is generated according to the current arm joint angles $q_{cur}$ and the expected arm movement $b_{i}$ in the *i*th step, which is limited to a small-scale σ.
(2)$$\begin{array}{cc} \Delta q & =f(q_{cur},b_{i})\\ b_{i}:\Delta p_{i} & =(\Delta x_{i},\Delta y_{i},\Delta z_{i})\\ s.t. & \sqrt{{\left(\Delta x_{i}\right)}^{2}+{\left(\Delta y_{i}\right)}^{2}+{\left(\Delta z_{i}\right)}^{2}}<\sigma \end{array}$$

Luo et al. proposed a method for robots to form the concept of direction developmentally [50] and generate basic unit movements according to the six directions: *up*, *down*, *left*, *right*, *forward*, and *backward*, with conditional generative adversarial networks [51]. However, this will lead to a non-smooth reaching trajectory in which the robot chooses to move in only one of the six directions at a time. We propose to generate the arm movement according to the relative position. The arm movement $b_{i}$ is a small-scale displacement toward the target, proportional to the relative position between the end-effector and the target with the amplitude under a given threshold σ. The choice of σ depends on the specific task requirements, robot dynamics, and safety considerations. By setting a small-scale displacement $b_{i}$, we ensure that the robot takes smaller, incremental steps during rough reaching to avoid potential collisions or unstable behavior. It also makes the reaching trajectory smooth. The threshold σ is larger than movement $b_{i}$ so that it will not reach the target in the first stage.

The end-effector of the robot arm can reach the target in infinite directions as long as the amplitude is under the threshold. With this method, we can generate a smooth trajectory, but the mapping space of the inverse model is too large to learn an accurate model. To narrow down the search space and speed up the planning, we propose to establish an inverse model for each joint of the robot arm. Then, the inverse model of the whole arm can be achieved by combining the single joints’ models. The output of an inverse model is the motion command of a corresponding single arm joint, so it is a one-to-one mapping and neural networks can be employed to characterize the inverse models.

As shown in Figure 2, for each of the *n* arm joints, a corresponding divided inverse model $DIM_{j}$ is built to generate a single motion command $\Delta q_{j}$ for that joint. The structure of an inverse model based on an artificial neural network is shown in the lower part of Figure 3. The input of the inverse model is the current joint angles and the expected movement, while the output is a motion command for the corresponding arm joint. The generated commands of different arm joints can be combined for whole-arm manipulation.

The forward model is a predictor that maps the robot arm joint angles to the spatial position of the end-effector. It can be used to predict the end-effector’s position before executing a motion command. The forward model is also a one-to-one mapping and is modeled with a forward neural network. The structure of the forward model is shown in the upper part of Figure 3. The input of the forward model is the arm joint angles and the output is the position of the end-effector.

#### 3.2.2. Two-Stage Learning for the Internal Models

Usually, we require a large amount of training data to train accurate models. However, it is difficult to collect data on a real robot because too many real executions will cause damage to the robot. To reduce mechanical wear and tear, a two-stage learning approach is proposed.

In the first stage, training data are collected in the simulation environment. A coarse traditional FK model is built based on manual measurements, which may not be accurate because of potential measurement errors. Restricted motor babbling is employed for self-exploration. At each time, only one joint of the arm moves within a certain range, while other joints are fixed. After motor babbling, the paired data of joint angles *q* and the positions of end-effectors *p* are collected and used to train a fine-tuned FK model. For each end-effector’s position $p_{i}$, an output joint angle $q_{i}$ is generated by the inverse model. We use the forward model as a supervisor, where the feedback is used to fine-tune the inverse model.

In the second stage, the internal models are optimized based on the execution of robot arm reaching in real robots. We collect training data with variable target positions and initial states of the robot arm in real robots. Visual feedback in the real environment is used to optimize the inverse model. A small amount of training data is sufficient in the optimization procedure.

The learning efficiency of internal models is improved through two-stage learning.

Studies on human arm reaching show that hand motion is learned from the error of distance and direction in the external coordinate system [52], which is the Cartesian space. Thus, during the training process of the inverse model, the loss is calculated as in Formula (Equation 3), i.e., the square error between the expected position of the end-effector and the position after executing the command.
(3)$$\begin{array}{c} L=||FM\left(q_{cur}+IM\left(q_{cur},b_{i}\right)\right)-\left(FM\left(q_{cur}\right)+b_{i}\right){\left|\right|}_{2} \end{array}$$

### 3.3. Motion Planning Based on Inner Rehearsal

As discussed before in Section 2, inner rehearsal can help robots to predict the result of a motion command without actual execution.

We propose a motion planning approach based on inner rehearsal for robot arm reaching. According to the relative distance between the target and the end-effector, the planning process is divided into two stages, proprioception-based rough reaching and visual-feedback-based iterative adjustments, as shown in Figure 4.

#### 3.3.1. Proprioception-Based Rough Reaching

In a rough reaching process, the end-effector is far from the target and it may not appear in the field of view of the robot. Thus, vision is only used to obtain the spatial position of the target $p_{target}$. The position of the end-effector $p_{hand}$ is derived from the forward model with the current joint angles $q_{t1}$ as the input at time t1. The expected arm movement $b_{i}$ is determined by the relative position Δp.

First, different motion commands are generated according to the different inverse models based on the current joint angles and the expected arm movement. Then, the forward model is employed to predict the results of these commands. An optimal motion command $\Delta {\hat{q}}_{t1}$ that brings the end-effector closest to the target will be chosen according to the greedy method. Instead of executing the command immediately, a sequence of motion commands will be generated through motion planning in an internal simulation. As Figure 4a shows, the inverse models will generate several motion commands based on the predicted position of the end-effector ${\hat{p}}_{hand}$ and the joint angles after executing the motion command $q_{t2}=q_{t1}+\Delta {\hat{q}}_{t1}$. Then, the forward model is used to select an optimal command $\Delta {\hat{q}}_{t2}$.

Repeating the above process, a sequence of motion commands $\left(\Delta {\hat{q}}_{t1},\Delta {\hat{q}}_{t2},\Delta {\hat{q}}_{t3},\ldots \right)$ to be executed will be generated based on inner rehearsal. The rough-reaching process will be finally realized by executing this motion command sequence.

#### 3.3.2. Visual-Feedback-Based Iterative Adjustments

This is a closed-loop control problem in the iterative adjustment process, with the visual information of the end-effector guiding the reaching process. Compared with the rough reaching process, the position of the end-effector is obtained from visual feedback rather than the output of the forward model. As shown in Figure 4b, the reaching method in this stage is very similar to the rough reaching method, except that, after a sequence of commands is generated, not all the commands but only the first few are executed. The closer to the target, the smaller the amplitude of the motion command should be. The detailed algorithm can be seen in Algorithm 1.
**Algorithm 1** Algorithm of motion planning based on inner rehearsal**Input:** Current joint states $q_{cur}$ and target position $p_{target}$**Output:** Motion commands Δq 1:$p_{cur}\leftarrow FM\left(q_{cur}\right)$ 2:${Seq}_{\Delta q}\leftarrow \Phi$ 3:k←0 4:**while** k<max_step **do** 5:   $\Delta p_{ori}\leftarrow p_{target}-p_{cur}$ 6:   **if** $||\Delta p_{ori}<\theta \left|\right|$ **then** 7:       Break 8:   **end if** 9:   Scale $\Delta p_{ori}$ into Δp so that Δp<σ 10:   **for** i=1→6 **do** 11:     $\Delta q_{i}\leftarrow {DIM}_{i}\left(q_{cur},\Delta p\right)$ 12:     $p_{i}\leftarrow FM\left(q_{cur}+\Delta q_{i}\right)$ 13:   **end for** 14:   Find the $p_{j}$ that is closest to $p_{target}$, ${Seq}_{\Delta q}={Seq}_{\Delta q}+\left\{\Delta q_{j}\right\}$ 15:   Update the state $\left[q_{cur},p_{cur}\right]\leftarrow \left[q_{cur}+q_{j},p_{j}\right]$ 16:**end while** 17:$\Delta q\leftarrow \Sigma {Seq}_{\Delta q}$ 18:Δq

## 4. Experiments

To evaluate the effectiveness of the proposed reaching approach based on inner rehearsal, several experiments are conducted. Some experimental settings and analyses of the results are introduced in this section. In the visual part, because it is not the key research part, we simply process the image in the HSV and depth space to extract the target information. It is encoded as a Cartesian (x,y,z) position.

### 4.1. Experimental Platforms

The proposed approach is verified on the Baxter robot and the humanoid robot PKU-HR6.0 II. To confirm the position of the end of the robot arm, we add a red mark to the end-effector and the robot detects its position throughout the experiment.

Baxter, shown in Figure 5a, is an industrial robot built by Rethink Robotics. It is a two-armed robot with 7 degrees of freedom (DoFs) for each arm, as shown in Figure 5b. In this work, we only use its left arm.

PKU-HR6.0 II is 58.56 cm tall and 4.23 kg in weight. There are 28 DoFs, including 2 in the head and 7 on each arm. Except for one DoF controlling hand grasping, the other 6 DoF ($RA_{1}∼RA_{6}$) on the right arm are shown in Figure 5c. PKU-HR6.0 II is equipped with an Intel RealSense SR300. The central controller of PKU-HR6.0 II is an Intel mini PC NUC517RYH.

The operating system is Ubuntu 14.04 LTS and ROS Indigo.

After verifying the efficiency and effectiveness of the inner-rehearsal-based robot arm reaching on the Baxter in a preliminary simulation, the PKU-HR6.0 II is used to test the performance of inner rehearsal in the real environment. To simplify the visual sensing and improve the recognition accuracy of the object and the end-effector, the target object is marked in green while the end-effector is marked in red, as shown in Figure 5d.

### 4.2. Evaluation of the Internal Models

#### 4.2.1. Model Parameters

The network parameters of the two platforms are shown in Table 2. In the simulation, the physical model is accurate, which means that we can directly use the chain rule to represent the forward model. Thus, the model fine-tuning procedure is not conducted on the Baxter robot. GlobalIM denotes the inverse model networks based on the accurate FK model, and DIM denotes the inverse model for joints.

In the PKU-HR6.0 II, FM denotes the forward model, and $DIM_{i}$ denotes the inverse model of the $i_{th}$ joint. The Adam optimization algorithm [53] is used to train each model. The output of the inverse model is a 6-dimensional motion command $\left(\Delta q_{1},\Delta q_{2},\ldots ,\Delta q_{6}\right)$, with all joint angles except the corresponding one set to 0.

#### 4.2.2. Data Preparation

For the Baxter robot, it obtains the training data through constrained DoF exploration. It moves to a fixed home position and then explores each DoF of the arm separately. For each DoF, it conducts 200 explorations within its range of angles. During this process, the position of the end-effector and the corresponding DoF states of each joint are recorded.

For the PKU-HR6.0 II, as the learning of the internal model is separated into two stages, data preparation is also divided into two stages. In the pre-training stage, 96,000 sets of original data (q,p) are generated through motor babbling to train the internal models. Adjacent data, where the amplitude is smaller than a predefined threshold σ = 0.5 cm, are processed to obtain the motion command Δq and its corresponding changes in the position of the end-effector Δp. Data that satisfy the conditions are selected for each inverse model $DIM_{i}$, where all except the *i*th component of Δq are zero. In total, 8000 sets of training data are selected for each inverse model.

In the fine-tuning stage, the robot executes 120 reaching processes. Approximately 3000 sets of data are collected for the forward model, 2400 for training, and 600 for validation. For each of the inverse models, 400 sets of training data and 100 sets of test data are collected.

#### 4.2.3. Performance of the Internal Models

In the real environment (PKU-HR6.0 II), the learning curves of the forward and inverse models in the fine-tuning stage are as shown in Figure 6a,b, respectively. It can be seen that the models converge quickly, only after a few epochs. Moreover, as stated in the previous section, we can see that, with training data generated from a coarse FK model, only a small amount of data in the real environment is required.

### 4.3. Evaluation of the Inner-Rehearsal-Based Motion Planning

In order to evaluate the effectiveness of inner rehearsal in improving the reaching accuracy, we conduct a set of comparative experiments. We compare the reaching performance between models with and without inner rehearsal.

One hundred sets of experiments are conducted with different target positions and different initial states of the robot in both platforms. The final distance between the end-effector position and the target position Δd is used to evaluate the proposed method. The reaching performance of each model is recorded.

For both platforms, orange represents the initial distance, while blue represents the distance after the reaching process. In the reaching process in simulation, the Baxter robot first performs coarse reaching according to the global inverse model, and then performs fine-tuning according to DIMs and the inner rehearsal results. The experimental results are shown in Figure 7a,b. As we can see from the result, with inner rehearsal, Δd has a significant decline.

In Section 3.3, we discuss how to plan robot arm reaching based on inner rehearsal. Here, the procedure without inner rehearsal differs in terms of how we choose the single joint movement at each time step. Without inner rehearsal, at each time step, we randomly choose one of the six inverse models and generate a motion command using DIM, with no prediction of the expected result. To evaluate the system quantitatively, for the final average distance between the target and the end-effector, we set 3.0 cm as the baseline approach without inner rehearsal. Meanwhile, the baseline is 1.2 cm for the proposed approach with inner-rehearsal-based motion planning, but without fine-tuning of the internal models. The baseline is 0.6 cm for the proposed approach with inner-rehearsal-based motion planning and model fine-tuning, as shown in Figure 8a,b,c, respectively.

The effectiveness of inner-rehearsal-based motion planning is verified by comparing Figure 8a,b. At the same time, the comparison result of Figure 8b,c shows the effectiveness of the learning approach for the internal models.

In the real environment (PKU-HR6.0 II), the motion trajectories of the two trials are also recorded and shown below. Figure 9a,b are the reaching trajectories generated from the approach without inner rehearsal. Figure 9c,d are the trajectories with inner-rehearsal-based motion planning but without model fine-tuning. Figure 9e,f are generated from the approach with motion planning and model fine-tuning. Comparing Figure 9a,c or Figure 9b,d, we can see that the motion trajectories are smoother with inner-rehearsal-based motion planning. Comparing Figure 9c,e or Figure 9d,f, it can be seen that the reaching performance based on fine-tuned internal models in the real environment is better. This experimental result demonstrates that our method of inner rehearsal can not only achieve smooth results in the trajectories, but also reduce the number of control points. This is because our method allows the robot arm to anticipate the next action when approaching, thereby improving the efficiency. Table 3 shows the average distance between the target and the end-effector before and after the reaching process in the two platforms. Due to the coarse forward model, our work focuses on improvement compared with the traditional method. Our purpose is to demonstrate the effectiveness of the human cognitive mechanism of inner rehearsal in improving motion planning. Due to fixed errors in mechanical equipment, it is difficult to completely reach the target.

## 5. Conclusions

In this paper, a robot arm reaching approach based on inner rehearsal is proposed. The internal models are pre-trained with a coarse FK model and fine-tuned in the real environment. The two-stage learning of the internal models helps to improve the learning efficiency and reduce mechanical wear and tear. The motion planning approach based on inner rehearsal improves the reaching performance by predicting the result of a motion command with the forward model. Based on the relative distance, the whole planning process is divided into proprioception-based rough reaching planning and visual-feedback-based iterative adjustment planning, which improves the reaching performance. The experimental results show that our method improves the effectiveness in robot arm reaching tasks. For the operation problem of the robotic arm, our work implements a relatively fixed two-stage framework. The human cognitive mechanism has great potential in enabling agents to learn to determine the strategic framework of grasping by themselves, so that robots can be more suitable for unknown, complex scenes. Furthermore, we can delve deeper into the cognitive mechanisms of humans and investigate how these insights can further enhance robot learning and decision making. In doing so, we can unlock new possibilities for robotic capabilities and their seamless integration into various applications.

## Figures and Tables

**Figure 1 biomimetics-08-00491-f001:**
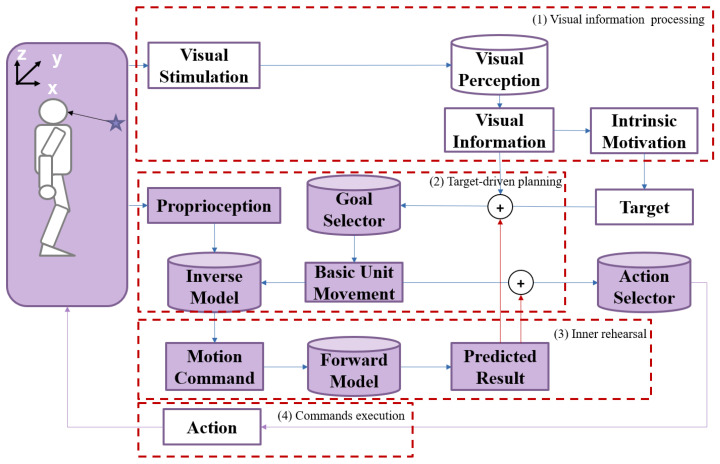
The overall framework of the robot arm reaching approach based on inner rehearsal. The purple shading in some boxes denotes “inner rehearsal”, different from “visual information processing”. The use of cylinders and rectangles is intended to represent different types of components in the system. Cylinders indicate “models”, while rectangles indicate “values”.

**Figure 2 biomimetics-08-00491-f002:**
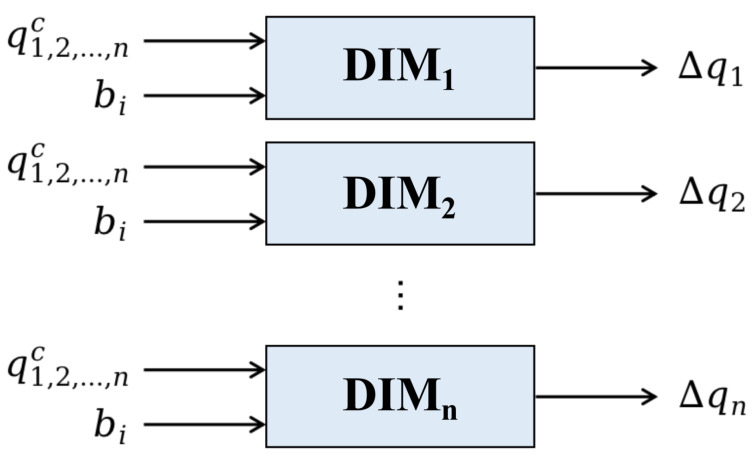
The inverse models for single arm joints of the robot.

**Figure 3 biomimetics-08-00491-f003:**
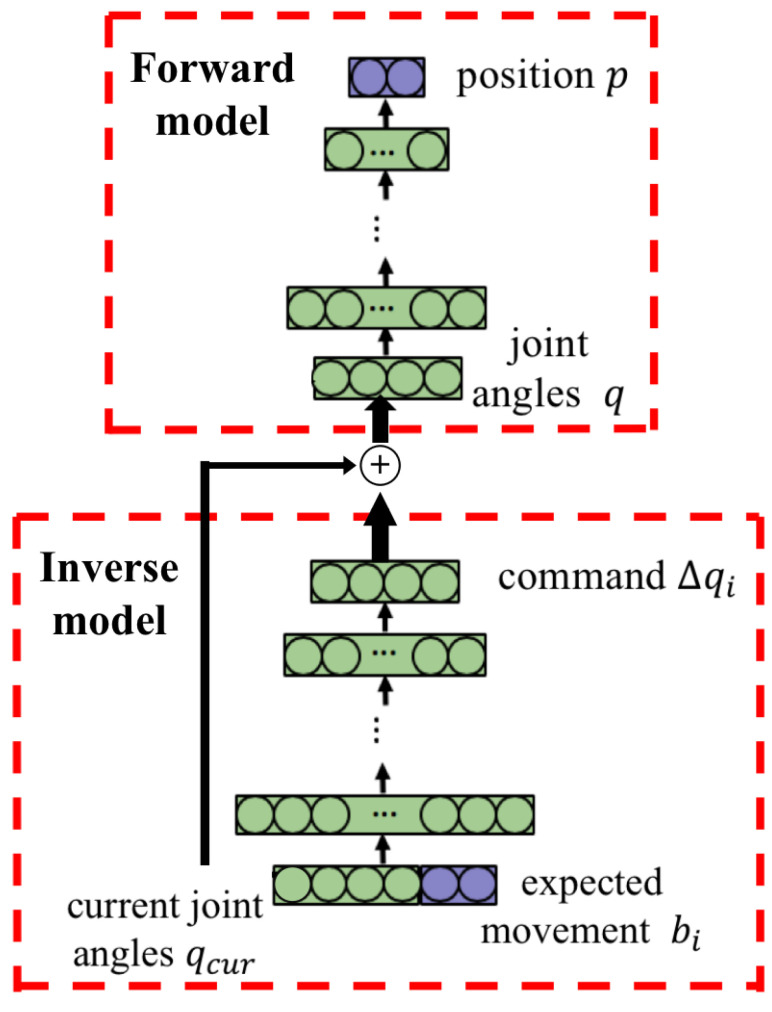
Structure of the internal models. The plus symbol inside the circle is the concatenation of vector summation. The forward model predicts the next time step’s Cartesian position given the sum of current joint angles $q_{cur}$ and joint angle command $\Delta q_{i}$.

**Figure 4 biomimetics-08-00491-f004:**
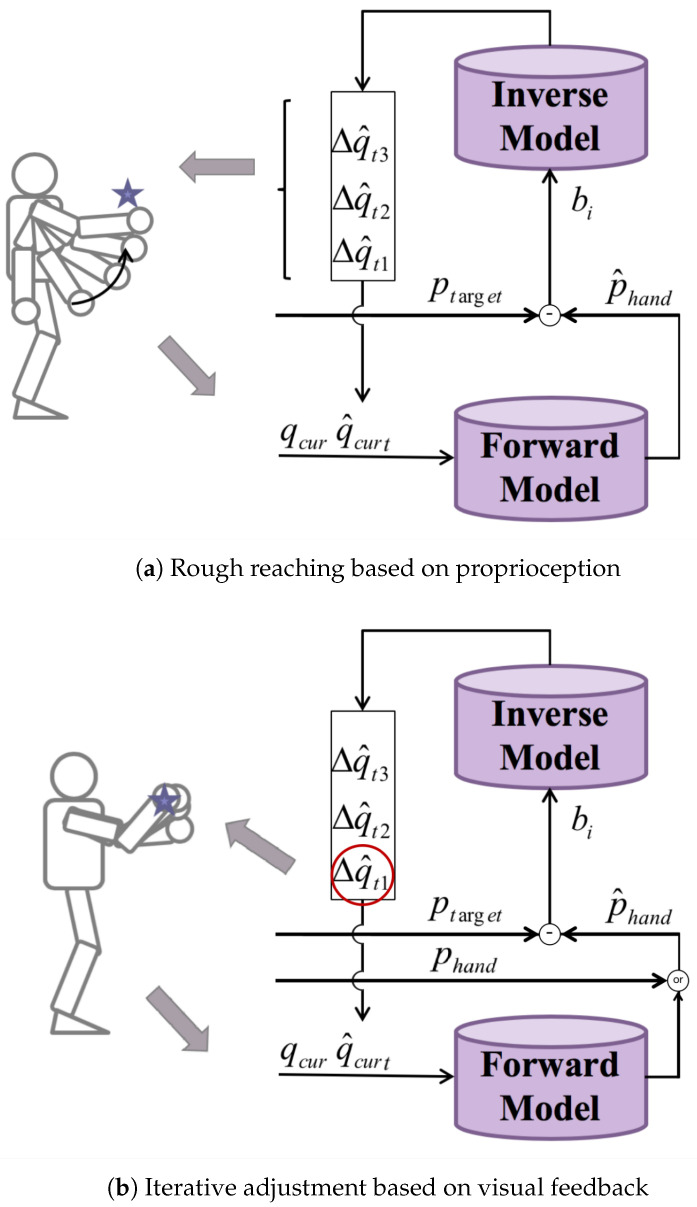
Motion planning approaches based on inner rehearsal.

**Figure 5 biomimetics-08-00491-f005:**
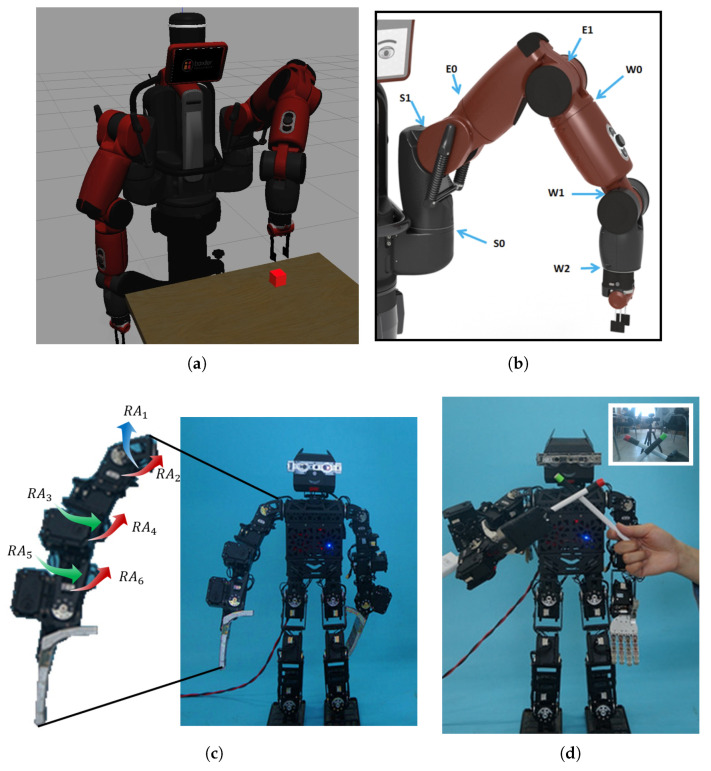
The experimental platforms, Baxter and PKU-HR6.0 II. (**a**) Baxter reaching for an object in a gazebo with its left arm. (**b**) Distribution of left arm joints in Baxter. The arm joints are named in the following manner: *shoulder roll (S0), shoulder pitch (S1), elbow roll (E0), elbow pitch (E1), wrist roll (W0), wrist pitch (W1), and wrist roll (W2)*. (**c**) The 6 joints on the right arm of PKU-HR6.0 II: *shoulder pitch, shoulder roll, elbow yaw, elbow roll, wrist yaw, and wrist roll*. (**d**) The object and the end-effector are marked in green and red, respectively.

**Figure 6 biomimetics-08-00491-f006:**
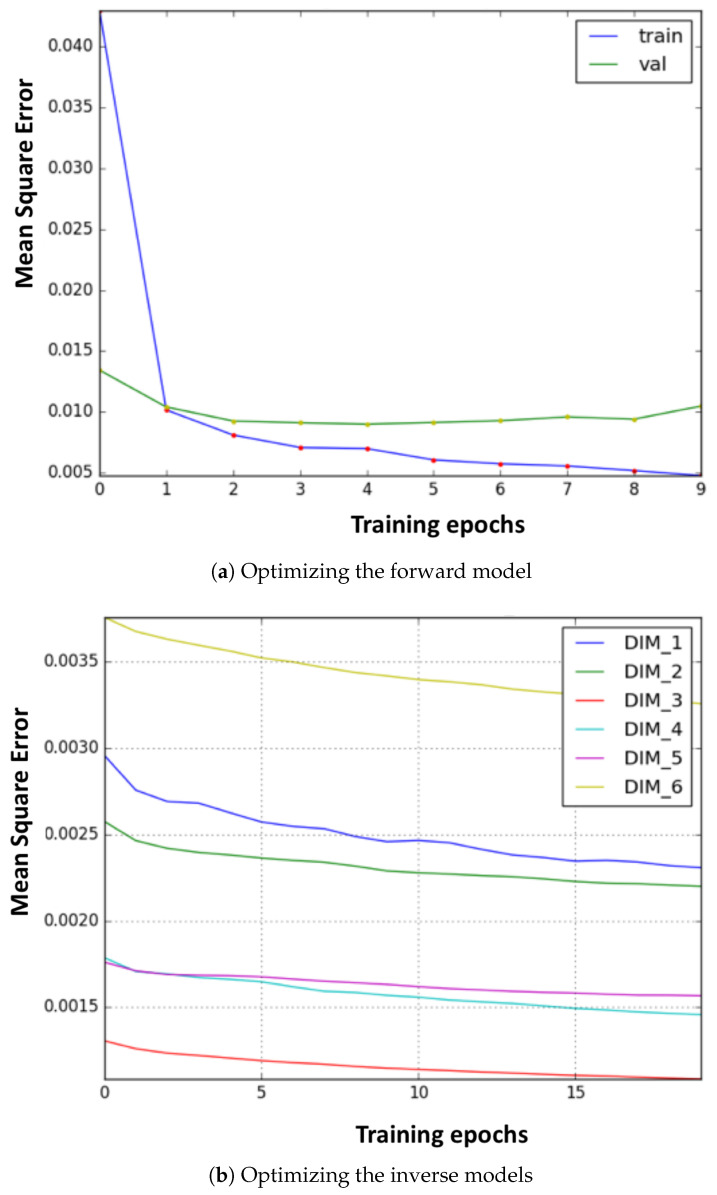
The optimization curve of the internal models. (**a**) Training process of the forward model. (**b**) Training process of the six inverse models.

**Figure 7 biomimetics-08-00491-f007:**
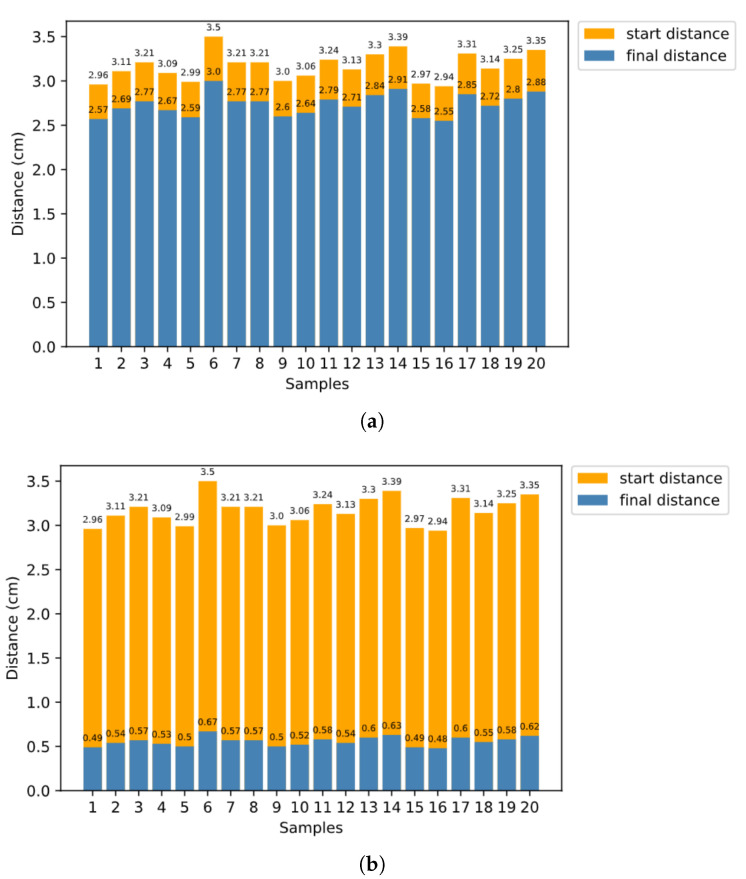
Distances between target and end-effector before and after the reaching process in real environment (Baxter). (**a**) Without inner rehearsal, (**b**) with inner rehearsal.

**Figure 8 biomimetics-08-00491-f008:**
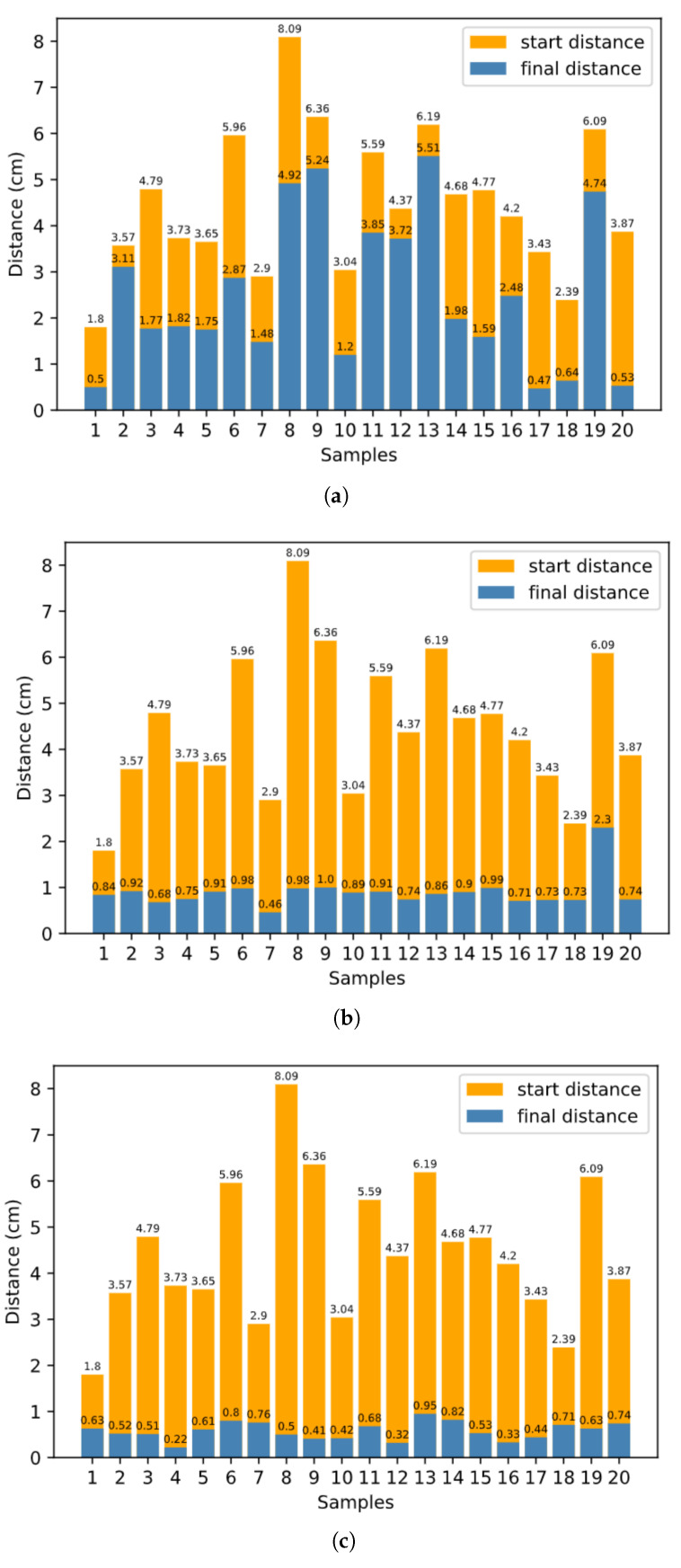
Distances between target and end-effector before and after the reaching process. (**a**) Without inner rehearsal, (**b**) with inner rehearsal but without model fine-tuning, (**c**) with inner-rehearsal-based motion planning and model fine-tuning.

**Figure 9 biomimetics-08-00491-f009:**
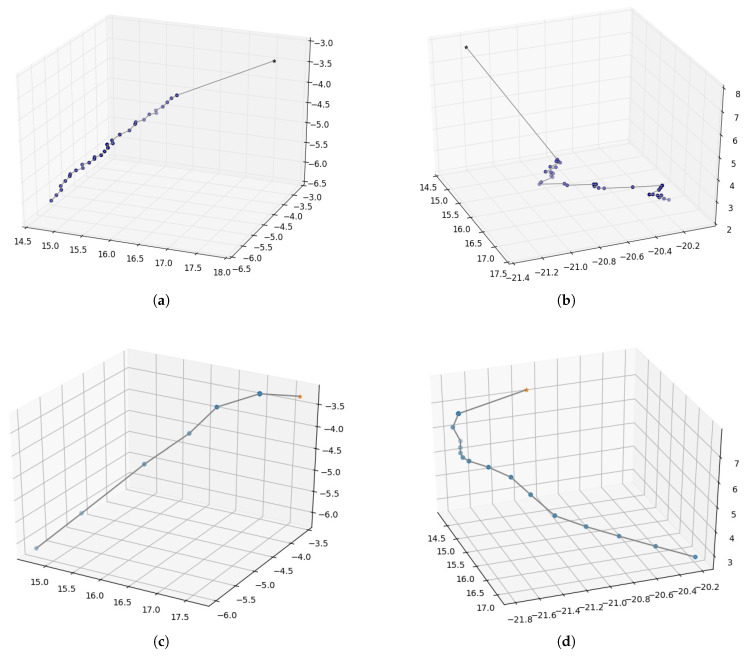
Motion trajectories of robot arm reaching. Pentagram represents the target position and the dots represent the positions of the end-effector after executing each motion command during the reaching process. (**a**,**b**) The trajectories generated by the approach without inner rehearsal; (**c**,**d**) the trajectories generated with inner rehearsal but without model fine-tuning; (**e**,**f**) the trajectories generated with inner rehearsal and model fine-tuning.

**Table 1 biomimetics-08-00491-t001:** Classification of robot arm reaching control approaches.

Classification	Approaches
	Numerical method [10]
Conventional IK-based	Analytical method [11,12]
	Geometric method [13]
Learning-based	Supervised learning: deep neural networks [14], spiking neural networks [15] Unsupervised learning: self-organizing maps [16], reinforcement learning [17,18]

**Table 2 biomimetics-08-00491-t002:** Network parameters of the internal models in Baxter and PKU-HR6.0 II.

Platform	Network	Size	Learning Rate
Baxter	GlobalIM *DIM*	3×64×64×7 3×64×64×1	0.0001 0.0001
PKU-HR6.0 II	FM	6×512×3	0.0001
	$DIM_{1}$	9×128×6	0.0001
	$DIM_{2,4,6}$	9×64×6	0.0001
	$DIM_{3,5}$	9×64×32×6	0.0002

**Table 3 biomimetics-08-00491-t003:** Average distances between the target and the end-effector before and after the reaching process in two platforms. The initial distances before reaching are 3.15 cm and 4.97 cm, respectively.

Platform	Condition	Distance after Reaching (cm)
Baxter	Without inner rehearsal	2.72 ± 0.19
	With inner rehearsal	0.55 ± 0.12
PKU-HR6.0 II	Without inner rehearsal	0.85 ± 0.16
	With inner rehearsal	0.53 ± 0.09

## Data Availability

Not applicable.

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
