# Peer review of "Robot Arm Reaching Based on Inner Rehearsal"

_biomimetics, 2023, doi:10.3390/biomimetics8060491_

Round 1

Reviewer 1 Report (Previous Reviewer 2)

The authors present arm reaching experiments by two robots, which are guided by trained forward- and inverse models. The authors' response to my previous review has helped clarification. I have a few remaining comments.

In Figure 3, the plus symbol inside the circle: clarify whether this is concatenation of vector summation. (This can be done in the caption or in the main text.)
In Figure 3, what does the forward model predict: (i) the Cartesian position at next time step, given current joint angles and commands, or (ii) the current Cartesian position given the current joint angles, or (iii) the next time step predicted Cartesian position given the sum of current joint angles plus commanded joint angles? (I guess the third. Please describe in the text.)

In Figure 4 a) and b), remove the top-most text "Rough reaching based on proprioception". It is redundant in a) and it is incorrect in b).

Page 10: you state that the image in HSV space is encoded as Cartesian (x,y,z)-position. However, this is hardly possible from a mono color camera. I assume you are using for both robots an RGB-D camera with depth information? Please clarify, e.g. write as "HSV+Depth space".

- introduce a space before citations [] and also after them if there is no comma or full stop

- page 3: "on robot arm reaching different platforms" -> "on robot arm reaching using different platforms"

- Figure 1 caption: use present tense "is" instead of "was"
  Correct to capital spelling of the word "Cylinders" after the full stop in the last line of the caption.

- page 14: "but also make the control points less" -> "but also reduce the number of control points"

- page 16: "our work has given a relatively fixed two-stage framework" -> "our work implements a relatively fixed two-stage framework"

Author Response

Reviewer 2 Report (New Reviewer)

In this paper, the authors presented an approach to robot arm reaching inspired by the human concept of inner rehearsal. The key idea was for the robot to predict or evaluate the outcome of a motion command before executing it. This approach enhanced the learning efficiency of models and reduced the mechanical wear on robots caused by excessive physical executions, as the authors claimed. The authors conducted experiments using the Baxter robot in simulation and the humanoid robots PKU-HR6.0 II in real/physical experiments to demonstrate the effectiveness and efficiency of the proposed approach for robot arm reaching across different platforms.

The objective seems to be good. The work presented herein does not prove an alternative to traditional inverse kinematics-based methods for robot arm reaching. The authors did not present any comparative results between these two concepts. The authors need to clearly justify how the proposed method can augment the effectiveness of inverse kinematics-based methods for robot arm reaching, or how the proposed method can replace inverse kinematics-based methods. Therefore, the novelty is not clear or justified.

The abstract of the manuscript must be improved. It should contain the research methods the authors used and the key results that the authors obtained. The results should be specific instead of some gross claims.

It is also not clear how significant the “inner rehearsal” of humans is to be transferred to robots, and how the developed robotic approaches were guided by inner rehearsal. It is at all not clear how the models proposed in Section 3 are justified through experiments and simulations in Section 4. The experimental results in Section 4 are very insignificant due to a single criterion (reaching distance) and do not justify the effectiveness of the proposed models.

Photos showing the experimental procedures should be presented. The future extension/direction of the research is not clear or adequate.

Minor editing of English language required.

Author Response

Reviewer 3 Report (New Reviewer)

I have the following comments:

1. in equation (1), what is m(t) exactly? Is it a vector or scalar?

2. what is s.t. in equation (2)?

3. A rigorous proof of convergence would be good.

Round 2

Reviewer 2 Report (New Reviewer)

The authors divided the previous review comments into 3 points, and provided their responses to each point. Their responses to point 3 are not at all satisfactory. Point 3 contains most of the issues the reviewer raised in the past review. However, the authors did not address this point seriously and adequately.

The revised submission does not show track changes or highlights for the revised content. Therefore, it is also not clear how the authors improved the manuscript based on the previous review comments.

Minor editing of English language required.

Author Response

This manuscript is a resubmission of an earlier submission. The following is a list of the peer review reports and author responses from that submission.

Round 1

Reviewer 1 Report

The paper presents a method for robotic arm reaching, where the robot's action is rehearsed internally before execution. Authors claim that the method enhances the learning efficiency of models and reduces the mechanical wear on robots caused by excessive physical executions. The method is demonstrated on two robotic (simulated Baxter and real PKU-HR6.0 II) platforms with and without internal rehearsal.

The motivation of the presented work is unappealing. Authors propose a machine learning (ML) based Inverse Kinematics (IK) alternative that performs worse (does not reach the desired end-effector state) than existing state-of-the-art IK solvers. Furthermore, the presented method does not do collision avoidance, self-collision avoidance, singularity avoidance, joint angle, velocity, acceleration, jerk, or torque constraints which are present in modern IK solvers. Authors are strongly advised to look into [1] - as an example of a modern real-time IK solver. 

The proposed method also depends on visual servoing which further reduces the applicability of the method.

"In the first stage, the models are pre-trained using a coarse forward kinematics model where the robot realizes self-exploration through motor babbling [47]." - forward kinematics of robotic arms (serial manipulators used as an example in the paper) can be solved efficiently analytically, why use less precise learning method? Authors are strongly advised to elaborate on why the learning approach is used instead of conventional analytical.

The experimental evaluation does not explain how the visual servoing is implemented. Authors are advised to describe the experimental setup in detail necessary to reproduce the experiment.

Authors are also advised to present information on task execution time. In general, the proposed method needs to be evaluated against existing works both learning and conventional IK solvers. Is the proposed method faster/slower than them? Is it more accurate? Does it adapt well to different unstructured environments, etc?

Experiment results show that the end-effector does not reach the target for both robots. Authors are advised to discuss this in more detail and how the method can be improved to actually reach the target. 

[1] D. Rakita, H. Shi, B. Mutlu and M. Gleicher, "CollisionIK: A Per-Instant Pose Optimization Method for Generating Robot Motions with Environment Collision Avoidance," 2021 IEEE International Conference on Robotics and Automation (ICRA), Xi'an, China, 2021, pp. 9995-10001, doi: 10.1109/ICRA48506.2021.9561505.

The paper is presentation is passable with good English and paper structure.

Reviewer 2 Report

The authors present arm reaching experiments by two robots, which are guided by trained forward- and inverse models. Some clarifications are still needed.

Please relate to famous old works by Wolpert and Kawato about inverse and forward models, e.g.: "Internal models in the cerebellum" (1998), or "Multiple paired forward and inverse models for motor control" (1998).

Considering the "small-scale displacement b_i" of the arm, how do you determine the threshold sigma? Can sigma be chosen large? (What prevents you from letting the robot move its arm very close to the target in a single step in the "rough reaching" phase?) This could be discussed under the background of the Equilibrium Point Hypothesis, e.g.:
Gu & Ballard (2006) An Equilibrium Point based Model Unifying Movement Control in Humanoids

line 273 "adding up these motion commands together and executing the combined one":
does this mean that one single (large) action command is executed, or a sequence of small motion commands? Clarify.

Figure 1: provide explanations: why do some boxes have purple shading, others not? What does it mean that some are drawn as cylinders, others as rectangles?

line 194: What does the index i mean in the expected arm movement b_i?

line 213, what does the abbreviation "DIM" stand for? Is it related to "IK" in equation 3?

Figure 2: Is it probably incorrect that all DIM instances have the same index "1".

Figure 3 appears misleading: shouldn't the input to the forward model be (i) the current joint angles and (ii) the command? Then, the layer that receives those two inputs should not be called "joint angles". Or is the input the sum of sum of current joint angles and command? Then, indicate this by a plus symbol. This would mean that the forward model is neither dependent on the current position nor on the action, but only on the desired position. So it merely transforms from angle space to (Cartesian?) position space? An equation, analogous to equation 2, but for the forward model, would help to clarify this.

Is "FK" in equation 3 the same as "FM" in Algorithm 1 line 12? Please be consistent with names (see also above, DIM and IK)

Figure 4 could be done better. How does information flow where lines are crossing? What does the red line segment in (b) mean? How can p_target and p_hand be written at the same line, although they mean something different?

Algorithm 1 line 10: What does the index i mean in the for loop?

Provide a brief explanation of the visual processing in the two robots, including how you get from the image to the "Target" (see figure 1). How is the Target encoded, is it Cartesian (x,y,z)-position?

Table 3, the distance before reaching is very small for the Baxter. Why is it not chosen larger? E.g. 50cm would easily be feasible and realistic in a typical real world setup. (Or are here the final positioning errors plotted before and after learning?)

Figure 9:the improvement from (a) to (b) shows the strength of your method. Write a paragraph intuitively explaining why the forward- and inverse models are necessary to yield this improvement. How comes that not only the trajectories become smoother, but also the control points (dots) become less?

Public release of the model code (at least for the part don in simulation) would be appreciated.

only minor flaws in English quality e.g.

l152 "to eliminates the influence"

l153 "The origin commands" -> "The original commands"

l176 "repeatly" -> "repeatedly"

Reviewer 3 Report

1) Expand on the keywords used in the article to provide a more complete representation of the topic. This can be achieved by including additional relevant terms and phrases that capture the different aspects and nuances of the topic under discussion.

2) Expand the references to previous work by the global scientific community (from 2020 to 2023) that has made significant contributions to the field. These references should include studies presented at prestigious conferences and published in top-tier journals such as Neurocomputing and other top-tier journals.

3) If possible, consider submitting any accompanying visual aids or diagrams in a vector format. The use of tools such as draw.io or similar software can help ensure that the visuals are of high quality and easy to edit, adding to the clarity and professionalism of the article.

4) In the conclusion, go beyond summarising the findings and implications of the current study. Discuss potential avenues for future research that could build on the current work and contribute to further advances in the field. Highlight areas that remain unexplored or unresolved, and suggest potential research directions and experiments that could be pursued.

5) While addressing any spelling and punctuation errors, pay attention to improving the overall style and coherence of the article. Refine sentence structure, eliminate redundant or awkward phrases, and ensure a consistent tone throughout the text. Although the article is generally easy to read, improving its linguistic quality will enhance its impact and readability.

Moderate editing of English language required.

Round 2

Reviewer 3 Report

The authors of the article carefully considered all the shortcomings and made significant improvements to their journal article.

Minor editing of English language required.